

# Spatial ecology of the Giant Sea Bass, *Stereolepis gigas*, in a southern California kelp forest as determined by acoustic telemetry

Kayla M. Blincow[1,2], Jack T. Elstner[2], Noah Ben-Aderet[2,3], Lyall F. Bellquist[2,4], Andrew P. Nosal[2,5] and Brice X. Semmens[2]

[1] Center for Marine and Environmental Studies, University of the Virgin Islands, St. Thomas, United States Virgin Islands, United States of America
[2] Scripps Institution of Oceanography, University of California, San Diego, La Jolla, California, United States of America
[3] California Department of Fish and Wildlife, San Diego, California, United States of America
[4] The Nature Conservancy, San Diego, California, United States of America
[5] Department of Biology, Point Loma Nazarene University, San Diego, California, United States of America

Corresponding author
Kayla M. Blincow,
kaylamblincow@gmail.com

## ABSTRACT

The fisheries history of the Giant Sea Bass, *Stereolepis gigas* (Telostei: Polyprionidae), is closely linked to its spatial ecology. Its overharvest is directly associated with formation of spatially distinct spawning aggregations during summer, while its subsequent recovery is hypothesized to be the result of spatially explicit gear restrictions. Understanding the spatial ecology of Giant Sea Bass is a key part of efforts to assess contemporary threats such as commercial harvest and incidental catch by recreational fisheries. In this study, we used acoustic telemetry to characterize Giant Sea Bass space use in the La Jolla kelp forest using an acoustic array that encompasses two marine protected areas (MPAs) and heavily trafficked recreational fishing grounds. Five of the seven fish we tagged remained in the La Jolla array for at least 6 months. Two fish were resident across multiple years, with one fish consistently detected for 4 years. Only one fish was detected in the broader network of regional acoustic receivers, moving north approximately 8 km to Del Mar. Most tagged fish had home ranges and core use areas indicating they spend considerable time outside MPAs, particularly in areas with high recreational fishing activity. During spawning season we detected fish less frequently in the La Jolla array and recorded higher movement rates. While the current MPA network in La Jolla by no means offers complete protection to this fish, it does appear to support long-term persistence of some individuals in a region of exceptionally high recreational fishing pressure.

## INTRODUCTION

Reaching over 2 m in length, the Giant Sea Bass, *Stereolepis gigas* (Teleostei: Polyprionidae), is one of the largest bony fish in the kelp forests off the coasts of southern

California and the Baja California Peninsula (*Hawk & Allen, 2014*). The Giant Sea Bass is a high-level predator that was once plentiful in coastal rocky reef habitats south of Point Conception, California (*Dayton et al., 1998*; *Domeier, 2001*; *Erauskin-Extramiana et al., 2017*; *Blincow et al., 2022*). Historically, it was a sought-after fisheries species, commercially and recreationally, which contributed to its near population extirpation from southern California waters (*Domeier, 2001*; *Baldwin & Keiser, 2008*; *Allen, 2017*). One contributing factor to its decline is its formation of spawning aggregations (*Allen, 2017*; *Erauskin-Extramiana et al., 2017*). This reproductive strategy can make fish easy to target once fishers identify an aggregation, because many individuals seasonally gather in the same geographic area (*Erauskin-Extramiana et al., 2017*). At the height of the Giant Sea Bass commercial and recreational fisheries in the US, fishers heavily targeted spawning aggregations during summer (*Allen, 2017*). The International Union for Conservation of Nature (IUCN) Red List of Threatened Species currently recognizes Giant Sea Bass as *Critically Endangered* (*Cornish, 2004*).

Recent reports indicate that the US Giant Sea Bass population is beginning to recover (*Pondella & Allen, 2008*; *Allen & Andrews, 2012*; *House, Clark & Allen, 2016*). In response to population declines in the early to mid-1900s, the state of California implemented regulations in 1981 that essentially closed all US Giant Sea Bass fisheries (FGC §8380, Title 14, CCR, §28.10). Currently, the government in California prohibits all recreational take of Giant Sea Bass, and commercial take in the state is limited to one incidentally caught fish per trip for gill net and trammel net fisheries (*Domeier, 2001*; *Baldwin & Keiser, 2008*). Reports of population recovery attribute the return of Giant Sea Bass to California waters to species-specific state fishing regulations, as well as the banning of the nearshore gill net fishery in 1994, which many believe reduced incidental landings (*Pondella & Allen, 2008*; *Allen & Andrews, 2012*; *House, Clark & Allen, 2016*; *Guerra et al., 2018*).

While reports of its recovery in the US are encouraging, the Giant Sea Bass still experiences fisheries take through the Mexican fishery, allowable commercial catch in the US, and incidental catch by US recreational fisheries. Recreational fishing of Giant Sea Bass in Mexico is limited to landing one fish per day (*Ramírez-Valdez et al., 2021*); however, there are currently no regulations on commercial Giant Sea Bass fisheries in Mexican waters (*Ramírez-Valdez et al., 2021*). It is difficult to gather reliable data on the status of the Giant Sea Bass fishery in Mexico because much of the catch is artisanal and often reported based on coarse regional areas or multi-specific groupings (*Erauskin-Extramiana et al., 2017*; *Ramírez-Valdez et al., 2021*). Fish production and consumptive value of Giant Sea Bass in Mexico are 19 times and 3.5 times greater than in the US, respectively (*Ramírez-Valdez et al., 2021*). If Giant Sea Bass travel between US and Mexican waters, the ongoing Mexican fisheries could be affecting populations managed by US agencies.

Fishing in the US could also be mediating the continued recovery of Giant Sea Bass. From 2000 to 2020, commercial fishers in the US landed an average of 2.76 metric tons of Giant Sea Bass per year (calculated from Pacific Fisheries Information Network (PacFIN) Commercial Landed Catch Species Report; www.psmfc.org). While this is much less than the landings reported prior to implementation of fishing regulations in California, it is still a large number of fish when considering this species' history of overfishing. Recreationally,

regulations limit Giant Sea Bass landings; however, a portion of individuals are caught incidentally and released. While recreational fishers are supposed to ensure the survival of incidentally caught Giant Sea Bass, it can be difficult to efficiently release fish of their size with barotrauma, especially if captured from larger vessels with raised decks (*Parker et al., 2006*). If not handled properly, barotrauma can result in fatality of the fish (*Parker et al., 2006*; *Jarvis & Lowe, 2008*).

The decline and subsequent rebound of Giant Sea Bass in US waters are linked to a complex history of spatial resource use and spatial management. From fishers actively targeting spawning aggregations (*Allen, 2017*; *Erauskin-Extramiana et al., 2017*), to the apparent positive response of US Giant Sea Bass populations to spatially explicit regulations limiting fishing gear types (*Pondella & Allen, 2008*; *House, Clark & Allen, 2016*), space appears to be an important consideration for conservation of this species. Acquiring a better understanding of how the species uses space can help determine the effectiveness of current management strategies and better understand the risks posed by contemporary fishing. For example, spatial management initiatives such as the California Marine Protected Area network, while not explicitly directed at conserving Giant Sea Bass, might provide benefits by protecting important habitat or providing refuge from fisheries.

The ongoing recovery of the Giant Sea Bass in southern California has allowed researchers to begin to consider its spatial ecology. As part of a larger regional multi-species mark-recapture study, *Hanan & Curry (2012)* recaptured two out of 14 tagged individuals 245 and 1,240 days post-tagging, one within 1 to 5 km and the other 5 to 20 km from the tagging locations. While only constituting data on two fish, this study suggests that Giant Sea Bass show some level of site fidelity. This finding is supported by more recent research tracking this fish on Santa Barbara Island, California, that found when 12 acoustically tagged individuals were queried across regional acoustic telemetry databases, they were detected solely on receivers stationed around the island, sometimes leaving the array but returning during spawning season (*Spector et al., 2022*). *Clevenstine & Lowe (2021)* used external acoustic tagging to investigate spawning aggregation site fidelity on Santa Catalina Island, California, and found tagged individuals resided at suspected spawning aggregation sites during the summer spawning season. About a third of the individuals tagged returned to the same spawning aggregation site in the subsequent year (*Clevenstine & Lowe, 2021*). However, they found that while some individuals remained on the island year-round, others traveled to other islands in the Channel Islands or the mainland coast of California (*Burns et al., 2020*; *Clevenstine & Lowe, 2021*). These excursions are a departure from the previous notion of Giant Sea Bass having limited home ranges (*Cornish, 2004*) and suggest that they can travel long distances.

In our study, we used acoustic tagging to characterize the spatial ecology of Giant Sea Bass individuals over a longer time scale than previously studied (>3 years), focusing on their movement in the La Jolla kelp forest. La Jolla, California, is one of the best areas for divers to observe adult and young-of-the-year Giant Sea Bass (*Allen, Benseman & Couffer, 2019*). The kelp forest overlaps with two separate no-take marine protected areas (MPAs) as well as one of the most intensely recreationally fished areas in the San Diego region (*Parnell et al., 2010*). Our objectives were to (1) determine whether tagged fish are resident

to La Jolla; (2) characterize the seasonality of space use; and (3) investigate how the movement of fish relates to spatial management and contemporary fishery-related threats.

## MATERIALS AND METHODS

### Study area

The La Jolla kelp forest (~8.25 km$^2$) is the second largest kelp forest in California (*Parnell et al., 2005*, *2006*) and is marked by hard bottom, with channels of sand and cobble interspersed throughout. On the northern edge, the La Jolla kelp forest is bounded by a submerged canyon with a sandy shelf and, on the western and southern edges, by sandy bottom habitats. In this area there are two no-take marine reserves, Matlahuayl State Marine Reserve and South La Jolla State Marine Reserve, as well as two conservation areas, San Diego-Scripps Coastal Marine Conservation Area and South La Jolla State Marine Conservation Area, which allow limited recreational and commercial fishing (Fig. 1). The region between these reserves is an important fishing ground for commercial sea urchin and spiny lobster fishers as well as recreational anglers from private vessels and the San Diego Commercial Passenger Fishing Vessel fleet, which are chartered vessels that take groups fishing (usually ~30–50 passengers) (*Parnell et al., 2010*). While our analysis focuses on the La Jolla region, we also shared our tag information with the network of researchers engaged in monitoring for acoustic tags in the broader region of southern California and Baja California, Mexico, and report those results as well (Fig. 1).

### Acoustic tagging

From August 2018 to October 2019, we tagged seven Giant Sea Bass in the La Jolla kelp forest (Fig. 1) using Vemco V16-4H acoustic tags (randomized 30 to 120 s reporting interval and 1,400 d battery life). These tags provide spatial and temporal presence information on individual fish. We monitored all tags from their date of release (Table 1) to 21 July 2022. We intended to have a larger fish sample size, but individuals were limited due to their rarity and our decision to fish <20 m. Ultimately, we only captured seven fish during 67 sampling days.

With the exception of one individual, we captured fish using hand lines with 9/0 or 10/0 circle hooks with whole dead Pacific Chub Mackerel (*Scomber japonicus*) as bait. We chummed the water using a combination of Shakin Bait (an Anchovy (*Engraulis mordax*) and Sardine (*Sardinops sagax*) based chum oil) and a frozen mixture of roughly chopped and/or blended Pacific Chub Mackerel, Pacific Jack Mackerel (*Trachurus symmetricus*), and/or Pacific Sardines. We targeted fish at depths <20 m and brought them to the surface at a moderate speed to minimize barotrauma while not exhausting the fish. To further address the potential negative effects of barotrauma, we reduced the time each fish spent at the surface. The only fish not captured using hand lines was caught with a gillnet during Hubbs Sea World Research Institute's (HSWRI) White Seabass (*Atractoscion nobilis*) survey conducted under contract for California Department of Fish and Wildlife (CDFW; Permit: P1770011) and approved by the HSWRI Institutional Animal Care and Use Committee (IACUC) (Protocol APF #2016-09). After being caught

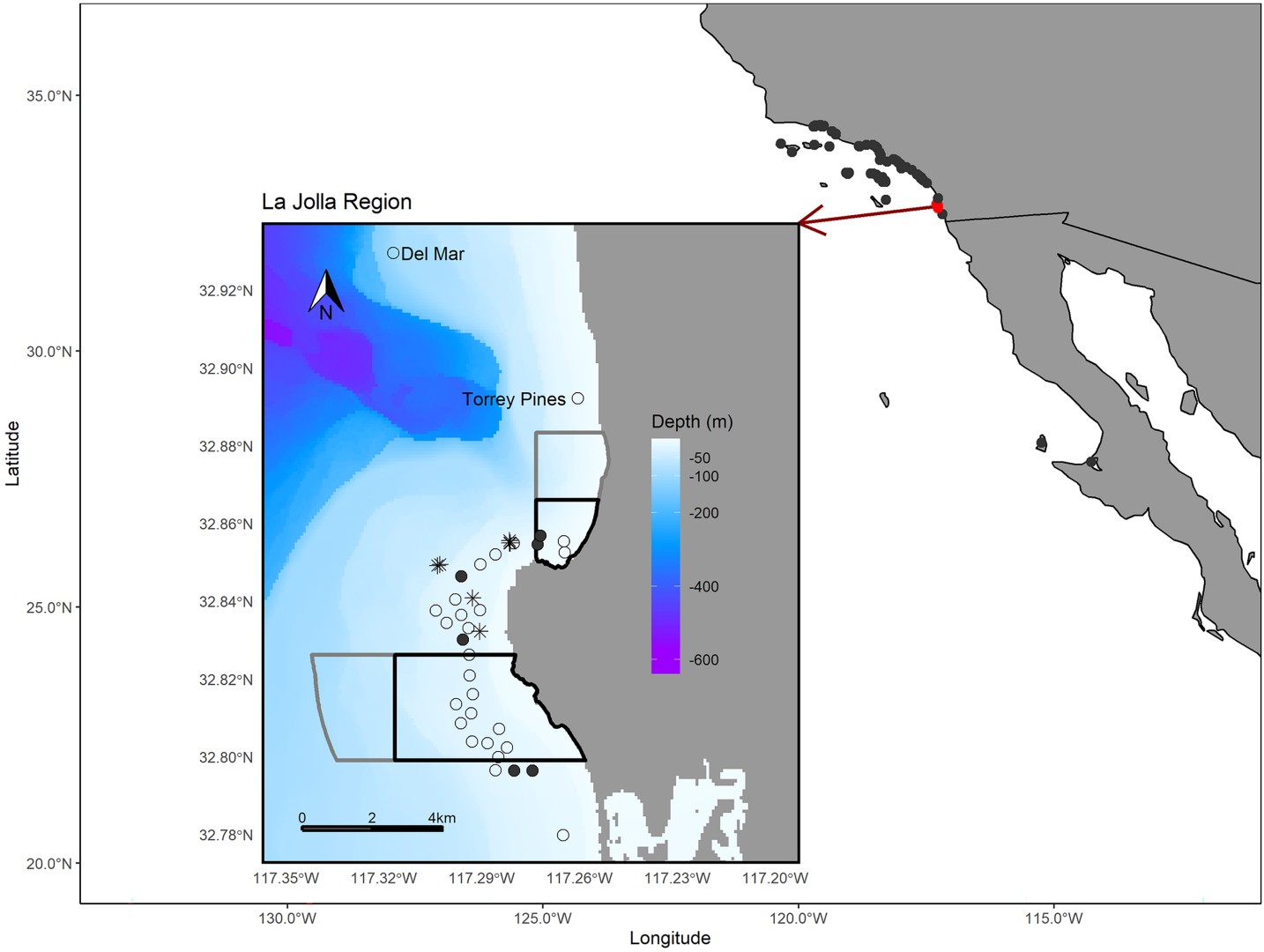

**Figure 1 Map of the study area.** The larger map depicts the array of regional receiver studies that we queried for detections of our tagged fish. The inset map shows receivers in the La Jolla region (the only receivers that detected our tagged fish). The black polygons depict the bounds of the Matlahuayl State Marine Reserve to the north and the South La Jolla State marine reserve to the south (*California Department of Fish and Wildlife, 2019*). The grey polygons depict the San Diego-Scripps Coastal Marine Conservation Area to the north and the South La Jolla State Marine Conservation Area to the south, which allow limited marine or recreational take (*California Department of Fish and Wildlife, 2019*). The open points denote the La Jolla array and our two receivers positioned to the north, Del Mar and Torrey Pines (labeled). Filled points depict the receivers where we performed range testing. The black asterisk symbols denote the locations where we captured Giant Sea Bass for tagging. The Pacific Ocean base layer used is publicly available via the San Diego Geographic Information Source (*Ross, 2018*) and the bathymetric layer is available via the National Centers for Environmental Information (*NOAA National Geophysical Data Center, 2003*).

in the gill net, we assessed if this fish was in good condition and transferred it to a holding tank before tagging.

We positioned fish captured using hand lines in a vinyl sling mounted on the side of our vessel to restrict movement while in the water. For the fish captured during the HSWRI survey (Tag 1), we kept it in a large, oxygenated holding tank before and after tagging. While we preferred to keep the fish submerged during tagging, we could not hold it steady at the surface in the holding tank or alongside the boat due to the fish's small size.

**Table 1 Summary data of tagged Giant Sea Bass, including tagging date, total length (TL) at tagging, and summary metrics of each fish's interaction with the La Jolla array.**

| Tag number | Tag date | Fish TL (cm) | Station count | Days at liberty (Study) | Days at liberty (Array) | Days detected | Study residency index | Array residency index |
|---|---|---|---|---|---|---|---|---|
| 1 | 8/15/2018 | 77 | 14 | 1,437 | 1426 | 425 | 0.296 | 0.298 |
| 2* | 11/9/2018 | 148 | 25 | 1,351 | 283 | 179 | 0.132 | 0.633 |
| 3 | 11/16/2018 | 117 | 21 | 1,344 | 864 | 767 | 0.571 | 0.888 |
| 4 | 7/22/2019 | 153 | 14 | 1,096 | 202 | 202 | 0.184 | 1.0 |
| 5** | 7/23/2019 | 163 | 0 | 1,095 | 0 | 0 | 0 | 0 |
| 6** | 7/24/2019 | 118 | 1 | 1,094 | 1 | 1 | 0.001 | 1.0 |
| 7 | 10/26/2019 | 107 | 22 | 1,000 | 271 | 187 | 0.187 | 0.690 |

Notes:
Study days at liberty refers to the number of days between the day after the release of the fish and the end of the study (21 July 2022) and array days at liberty refers to the number of days between the first and last detection of the tag in the La Jolla array.

* This fish left the La Jolla array and traveled north to the Del Mar receiver.

** These fish either had no detections after initial data filtering (Tag Number 5) or only had detections from two receivers the day after tagging (Tag Number 6), so were removed from subsequent analyses.

We removed the fish from the holding tank and placed it in a vinyl cradle on the deck of the vessel during surgical tagging. We covered the fish in a wet towel and used a seawater hose to maintain water flow over its gills for the short period spent outside of the holding tank (<2 mins).We implanted acoustic tags in each fish's gut cavity *via* an incision off-center of the midline and posterior to the pelvic girdle following *Lowerre-Barbieri et al. (2014)* and *Blincow et al. (2020)*. We used sterile antibiotic-infused, dissolvable cutting sutures ("PDS II violet 27" CP-1) to close the incision. We measured each fish for total length (TL; cm), standard length (SL; cm), and head length (HL; cm). We secured an external Floy tag (BFIM-96) at the base of the dorsal fin as a visual identifier of surgically tagged fish. Later, we positioned the fish to recover in a dorsal side-up position alongside the vessel (or within the holding tank for the HSWRI fish) before release. If fish had inflated swim bladders, we released them at depth using a descending device (SeaQualizer). CDFW permitted our activities (Permit #S-192900002-19290-001), and the University of California, San Diego IACUC approved our tagging protocols (Protocol #S12116).

## Acoustic receiver arrays

We used a stationary receiver array comprised of 29 Vemco VR2W single channel passive autonomous data-loggers, deployed in the La Jolla kelp forest to track tagged fish movements. Each VR2W receiver logged date, time, and individual. The depth range of receivers was 11.27 to 24.69 m (19.81 m ± 3.04; Mean ± SD). In addition to the 29 receivers moored in the La Jolla kelp forest, we had two receivers moored to the north at Torrey Pines (VR2W) and Del Mar (VR2C) (Fig. 1). The Torrey Pines receiver was deployed adjacent to a ~1 acre artificial reef habitat constructed by CDFW in 1975 of quarry rock and concrete dock floats at a depth of ~13 m as part of their Nearshore Sportfish Habitat Enhancement Program (*Lewis & McKee, 1989*). The Del Mar receiver was placed at 13 m depth on a multidisciplinary surface mooring deployed on the shelf break at 100 m just offshore from rocky reef, kelp forest habitat along the coast (*Send & Name, 2012*; *Navarro, Parnell & Levin, 2018*). We checked for detections of our tagged fish by other regional

acoustic receiver arrays ranging from Isla de Cedros, Baja California, Mexico, to Santa Barbara, California, USA (Fig. 1). These arrays were active throughout the study period.

We performed a detection range analysis on six of the 29 receivers in the La Jolla array (*Blincow et al., 2020*) (Fig. 1). The six receivers chosen spanned a representative depth gradient for the array, ranging from 15.54 to 24.38 m (20.12 m ± 3.04). We performed drifts starting at the coordinates of a given receiver mooring while towing a Vemco-coded transmitter tag (~1 to 2 m depth). We simultaneously recorded all acoustic tag transmissions (pings) during the drift using a Vemco VR100 mobile receiver unit deployed off the vessel in close proximity to the tag and captured detections with those recorded on the moored VR2W receivers. Using the coordinates for each ping detection on the VR100, we calculated the distance of each ping from the VR2W receiver mooring. We compiled data for all receivers that detected the towed tag and analyzed them using a generalized linear mixed-effects model (glmm) with a logit link and a random slope effect of receiver to determine the detection probability of individual pings (binary response) and distance of the tag from the receiver (continuous covariate). With the exception of our movement rate analysis (described below), we assumed the detection range of all of our receivers to be the distance at which our model estimated we could detect tag pings with a 50% probability. We note that detection ranges can vary depending on environmental factors, such as diurnal noise patterns and current variability (*Mathies et al., 2014*; *Huveneers et al., 2015*); however, we chose to make the simplifying assumption of a relatively constant detection range over time for all our receivers.

## Data analysis

We estimated the age of tagged fish based on their sizes and available age-growth relationships for Giant Sea Bass (*Hawk & Allen, 2014*).

Prior to analysis, we filtered data to remove detections on the same day as when we tagged the fish to avoid any behavior associated with recovery from tagging influencing our results (*Farmer & Ault, 2011*). To avoid spurious detections from code collisions, we removed any detections from the same tag on a single receiver across time intervals that were less than the minimum time it takes the tag to transmit a signal. We performed all analyses using R statistical software, version 4.1.1 (*R Core Team, 2019*). We implemented our models using a maximum likelihood approach with the 'lme4' package (*Bates et al., 2015b*) and estimated associated $p$ values using the 'lmerTest' package, which uses the Satterthwaite approximation of degrees of freedom method (*Kuznetsova, Brockhoff & Christensen, 2017*). This method estimates the denominator degrees of freedom for F statistics or degrees of freedom for $t$ statistics, depending on the model structure, to evaluate significance and produces more conservative $p$-value estimates with lower levels of Type one error rates when compared to other mixed-effect model $p$-value estimation methods, such as likelihood ratio tests (*Luke, 2017*).

We calculated the number of days each fish was at liberty across the study period by determining the number of days between the day after the release of the fish and the end of the study (21 July 2022). We also calculated the number of days each fish was at liberty within the La Jolla array by determining the number of days between the first and last

detection of the tag on La Jolla array receivers after data filtering. We calculated an array residency index, or the residency of each tagged fish within the La Jolla array, by dividing the number of days each fish was detected within the array by the number of days they were at liberty within the array. We calculated a study residency index, or the residency of each tagged fish within the La Jolla array across the entire study period, by dividing the number of days each fish was detected within the array by the total number of days they were at liberty across the study period. One of our fish left the La Jolla array and was detected consistently at the Del Mar receiver for a period of months.

We summarized fish movements in La Jolla by calculating their activity spaces within the La Jolla array. First, we generated position estimates by calculating centers of activity (COAs), which are weighted average positions of each fish based on the number of detections present on each receiver across 30 min intervals (*Simpfendorfer, Heupel & Hueter, 2002*). We then used these COAs to calculate the 50% and 95% kernel utilization distributions (KUD) of each fish across spawning and non-spawning seasons in the La Jolla array using the 'adehabitatHR' package in R (*Calenge, 2006*). We used the *ad hoc* method for determining the smoothing parameter for KUD calculations, which assumes the utilization distribution is bivariate normal (*Calenge, 2006*). We limited our KUD analyses to the La Jolla array due to our interest in investigating interactions between Giant Sea Bass resident to La Jolla with local spatial management. There was only one fish detected outside of the La Jolla array, and it appeared to emigrate from La Jolla to the vicinity of the Del Mar receiver (the fish spent 9 months in La Jolla and then traveled to Del Mar and never returned; Fig. 2). If this fish had made regular excursions to Del Mar and back to La Jolla, we would have considered the full range of its movements in the KUD analysis. The 50% KUD is representative of the core use area of the fish, while the 95% KUD is representative of the home range of each fish within the La Jolla array. There was one instance in which the 95% KUD overlapped with land along the coast. In this case, we removed the land portion of the KUD. We used the resulting KUD estimates to calculate the area of overlap between La Jolla MPAs with core use areas and home ranges.

To investigate seasonal and diel differences in the activity of our fish in La Jolla we calculated hourly movement rates, which we defined as the distance moved during 1-h intervals. We did this by estimating COAs as described above across 10 min intervals. Since we are unable to measure movement rates when the fish are outside of the detection range of the array, we filtered for intervals consisting of six consecutive COAs. We summed the distance between COAs in the resulting hour intervals to generate hourly movement rates when fish were occupying the La Jolla array. The movement rate data were zero-inflated, so we analyzed them using two separate models. First, we converted the movement rates to a binary variable, with 0 being a zero movement rate and one being a non-zero movement rate. Using this information, we constructed a binomial glmm (logit link function) to calculate the probability of a positive movement rate given the explanatory variables of diel period (dawn, day, dusk, or night), lunar phase (waxing, full, waning, new), and month. Second, we filtered our data for only non-zero movement rates

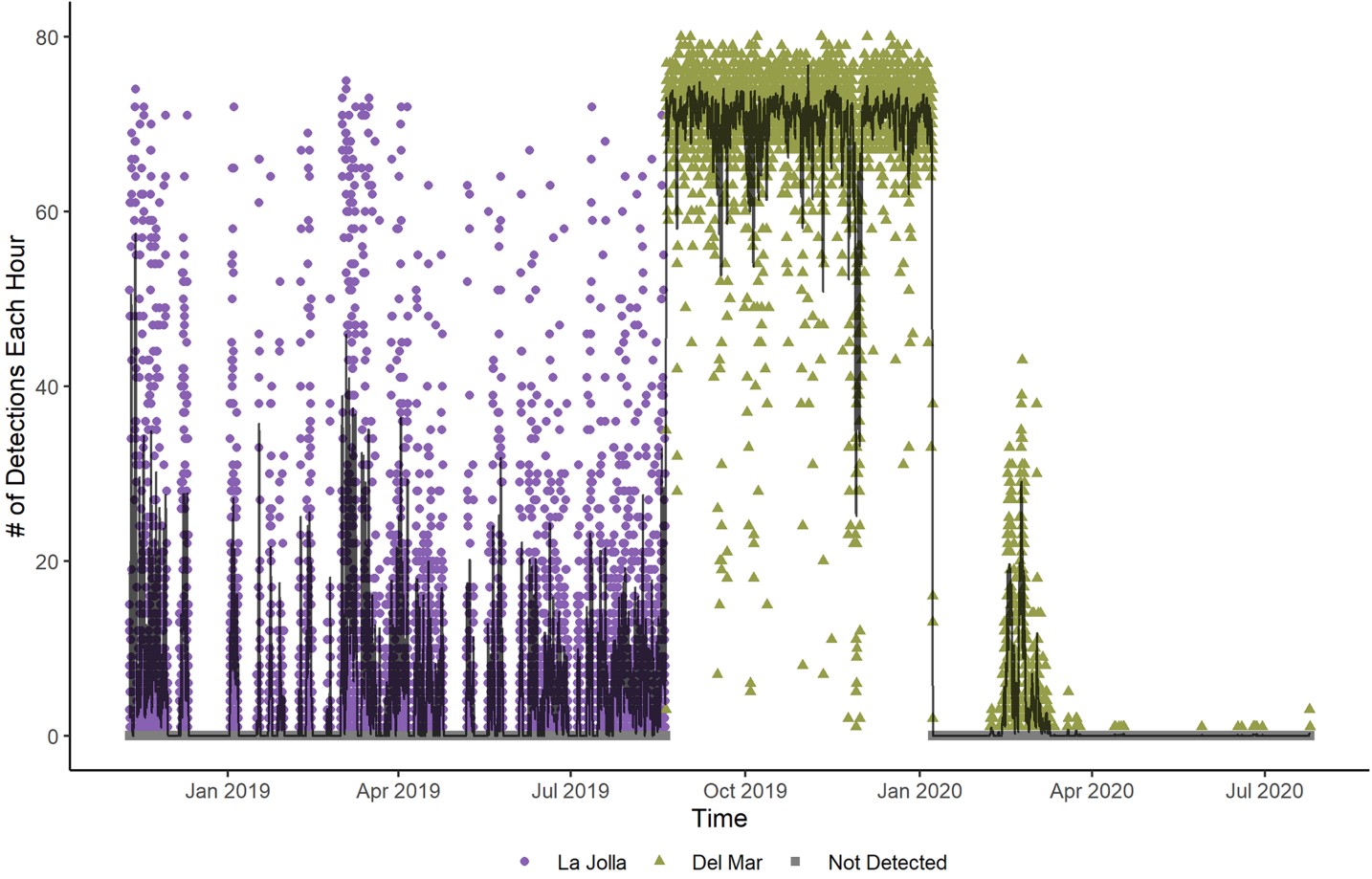

**Figure 2 Number of detections each hour for Tag Number 2 in the La Jolla array (purple circles) and at the Del Mar Receiver (green triangles).** The line shows the moving average of the hourly detections across 12 h. The fluctuations shown in this plot indicate that the fish is alive and active, despite the increased frequency of detections while at the Del Mar receiver.

and used a linear mixed-effects model to determine the effect of the explanatory variables diel period, lunar phase, and month. In accordance with suggested best practices for mixed effects modeling that recommend fitting the most complex mixed effects structure allowed by your data (*Bates et al., 2015a*; *Harrison et al., 2018*), we first attempted to fit models with both random slope and intercept terms; however, we were unable to reach model convergence. As a result, both of our models included only a random intercept effect of individual, the most complex model structure allowed by our data. We calculated the associated pseudo-$R^2$ values (marginal and conditional) using the delta method *via* the 'MuMIn' package (*Nakagawa, Johnson & Schielzeth, 2017*; *Barton, 2022*).

## RESULTS

We tagged seven fish (126.14 ± 30.25; Mean ± SD; range 77 to 163 cm TL) (Table 1). Based on their sizes and published age-growth relationships, all the fish we tagged were likely sexually mature; however, our smallest individual fell within the range of uncertainty regarding age at maturity for the species (Tag 1 estimated age: 9 years, species reported age at maturity: 7 to 13 years). The number of days each tagged fish was at liberty throughout

the study ranged from 1,000 to 1,437 (1,202.43 ± 169.68) (Table 1). The number of days each fish was at liberty within the La Jolla array ranged from 0 to 1,426 (435.29 ± 524.09) and the number of receivers each fish was detected at ranged from 0 to 25 (13.86 ± 9.99) (Table 1). The La Jolla array residency index ranged from 0 to 1 (0.64 ± 0.38) while study residency index ranged from 0 to 0.571 (0.20 ± 0.20) (Table 1). Two fish (Tag Numbers 5 and 6) left the La Jolla array within two days of being tagged and did not return (Table 1). We removed these fish from subsequent analyses.

VR2W receivers in the La Jolla array on average detect tag pings with a 50% probability at 218.3 m (Fig. 3). Receivers in sandy areas on the edges of the kelp forest had a larger detection radius than receivers within the kelp forest. The largest distance of 50% detection probability calculated for an individual receiver was 283 m and was associated with a receiver moored at 24.38 m depth in the open sandy area between the edge of the submarine canyon and kelp forest on the northwest edge of the array. The lowest was 180 m and was associated with a receiver moored at 17.07 m depth in the kelp. The presence of kelp (or lack thereof) appeared to outweigh other factors that could potentially influence detection ranges, including presence of currents or depth.

Two fish (Tag Numbers 1 and 3) remained within the La Jolla array consistently throughout their time at liberty, a period of 2.37 and 3.92 years respectively. Three fish (Tag Numbers 2, 4, and 7) left the array bounds after approximately 9, 8, and 6 months, respectively (Fig. 4A). One of these three fish traveled to the Del Mar receiver (movement rate: ~0.45 m/s) and remained there consistently for approximately 5 months before leaving and returning again to Del Mar (Fig. 2). Our fish were not detected at any other receivers from the broader southern California and Baja California, Mexico regional arrays.

We found that KUDs varied across fish, but that the area between the two MPAs was the most highly used area overall (Fig. 4). During non-spawning season, the 95% KUDs for tagged fish had, on average, 32% overlap with local MPAs, while during spawning season this overlap grew slightly to 36%, with a larger proportion attributed to space use within the Matlahuayl State Marine Reserve (Fig. 4). The core use areas (50% KUD) of all fish averaged 9% overlap with MPAs in non-spawning season and grew to 20% overlap during spawning season; this latter finding was predominantly driven by the fish with Tag Number 7 (Fig. 4).

Based on our movement rate analysis, fish had a lower probability of non-zero movement rates during summer months, particularly June through September (Table 2, Fig. 5B). Given a positive movement rate, the predicted movement rates were highest in the months of May through July (Table 3, Fig. 5E). These periods coincide with the recorded spawning months of Giant Sea Bass (May through October). We recorded fewer detections per hour as well (Fig. 5A). The probability of non-zero movement rates was highest during the day, though all diel periods had predicted probabilities of non-zero movement rates inclusive of confidence intervals that were greater than 35% (Table 2, Fig. 5D). Given

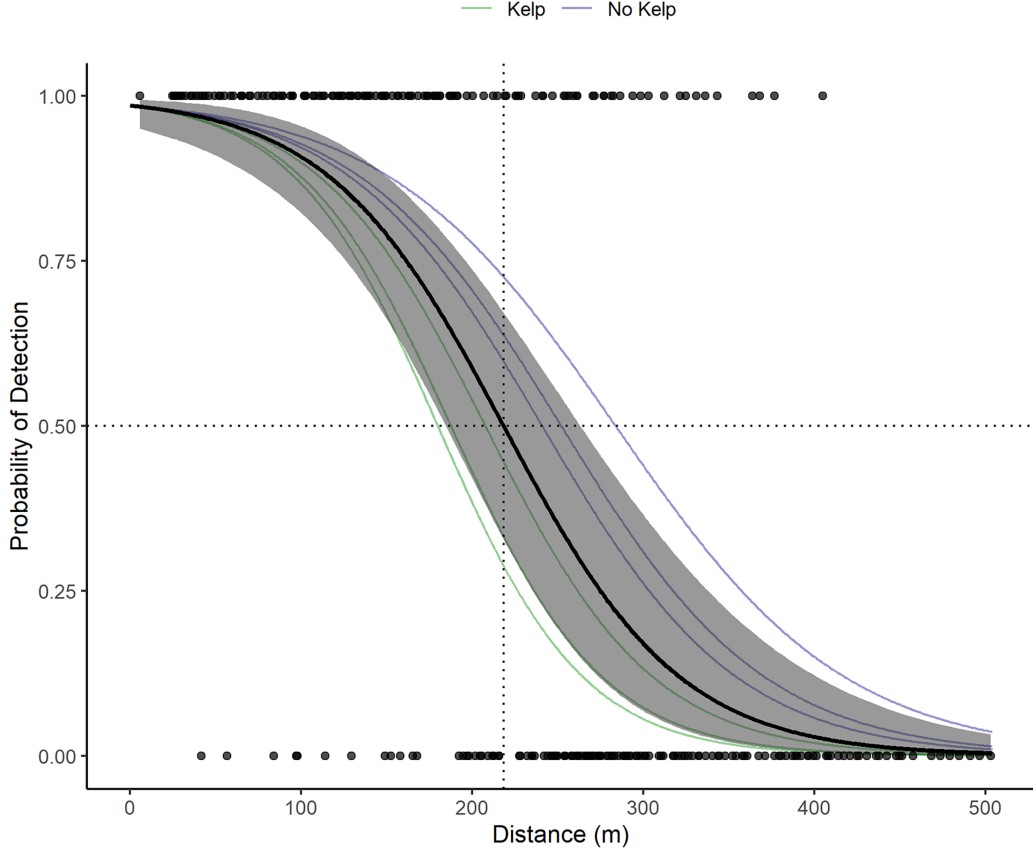

**Figure 3 Acoustic receiver range testing results.** The points depict the binary detections of the receivers range tested (1 for pings detected on both the VR100 and the range tested receiver or 0 for pings only detected on the VR100). The solid black line and gray show the global mean estimate and associated 95% confidence interval for the probability of detection with distance of all receivers. The dotted black lines show where the global mean estimate has a 50% probability of detection (218.3 m). The faint green and blue lines depict the random effects estimates for each receiver and are color coded based on whether the receiver was located in kelp habitat or not.

non-zero movement rates, the predicted movement rate was lowest during nighttime and higher across all other diel periods (Table 3, Fig. 5F). The predicted movement rate did not differ between dawn, day, or dusk (Table 3, Fig. 5F). We did not find a relationship between lunar phase with either the probability or rate of movement (Tables 2, 3). Both movement models had relatively low conditional $R^2$ values (binary movement: 0.141, non-zero movement: 0.051), suggesting that there is a large amount of variability in the data that is unaccounted for by the explanatory variables included in the models (Tables 2, 3).

## DISCUSSION

We found that some Giant Sea Bass are long term residents of the La Jolla kelp forest. Fish had the highest predicted movement rates during summer spawning months and tended to be detected more outside of the La Jolla array during this same time period, suggesting that a local spawning aggregation site exists outside the bounds of our array. When fish were

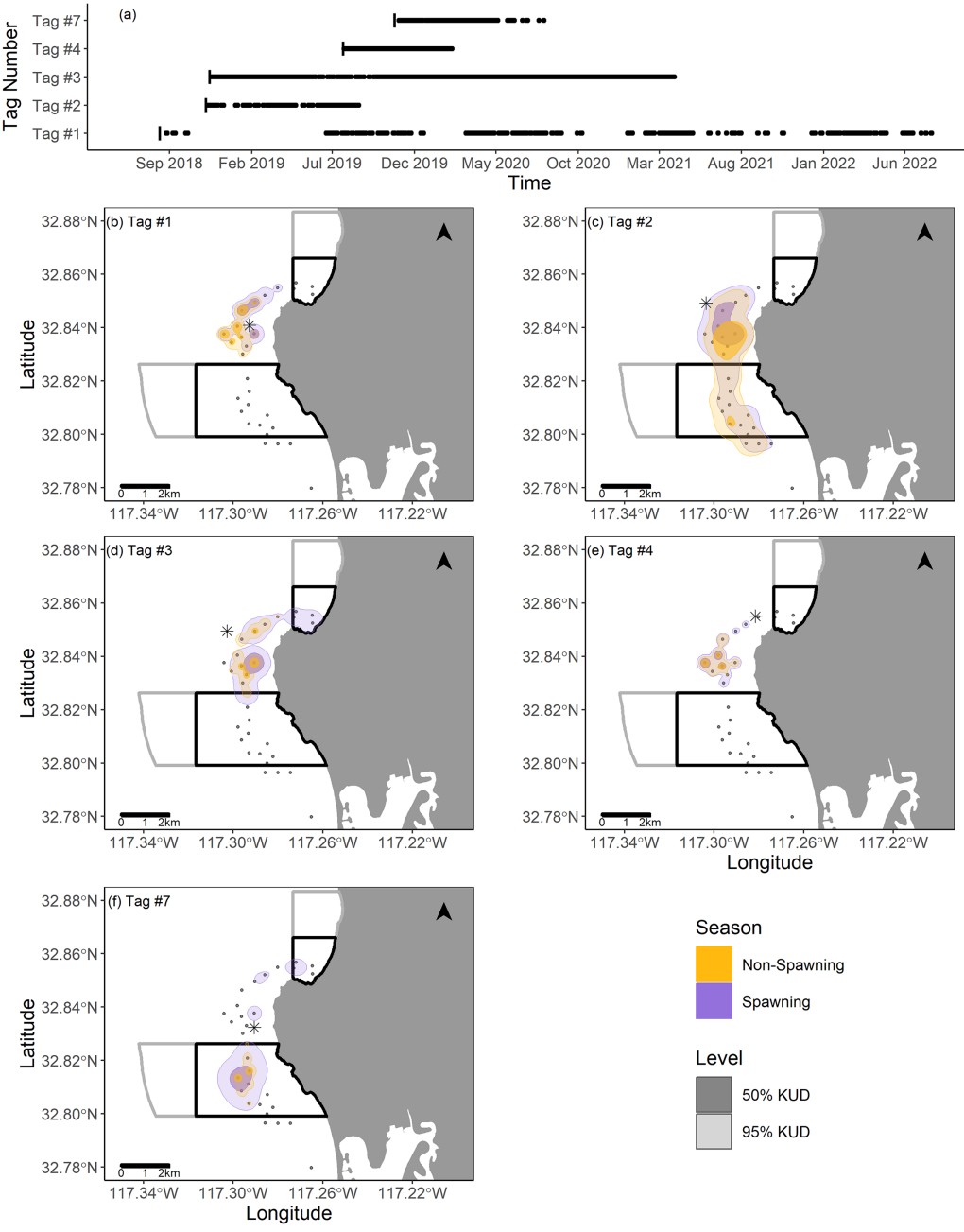

**Figure 4 Summary of detection data and kernel utilization distributions (KUDs) for the five tagged fish used in our analysis.** (A) Detections within the La Jolla array across time. (B–F) 95% and 50% KUDs for spawning and non-spawning seasons overlayed on the La Jolla array for each fish (tag number specified in the upper left corner). The asterisk points denote the capture location for each fish, the black boxes depict the no-take marine protected areas (*California Department of Fish and Wildlife, 2019*), and the gray boxes depict conservation areas that allow limited take (*California Department of Fish and Wildlife, 2019*). The Pacific Ocean base layer used is publicly available *via* the San Diego Geographic Information Source (*Ross, 2018*).

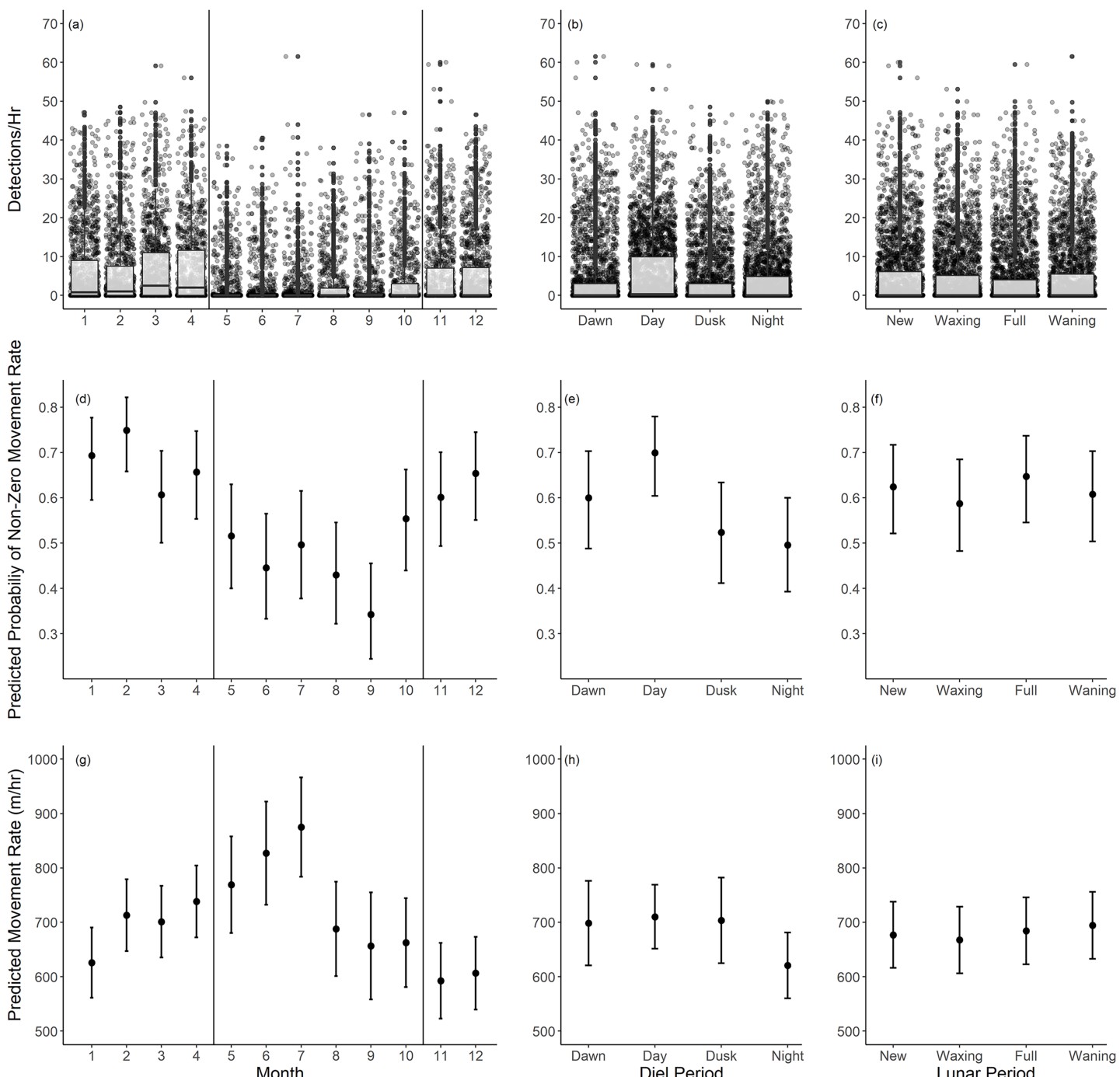

**Figure 5 Summary of monthly, diel, and lunar movement patterns of Giant Sea Bass in La Jolla.** (A–C) Box plots depicting the detections per hour for each tagged fish across months (A), diel periods (B), and lunar phases (C). (D–F) The predicted probability of non-zero movement rates and associated 95% confidence intervals across months (D), diel periods (E), and lunar phases (F) based on our binomial generalized linear mixed-effects model. (G–I) The predicted rate of movement and associated 95% confidence intervals given a non-zero movement rate across months (G), diel periods (H), and lunar phases (I) based on our linear mixed-effects model. Note there was not a significant difference in probability or rate of movement across lunar phases. The black vertical lines in the left panels show the range of spawning months reported for Giant Sea Bass.

**Table 2 Summary of the binomial generalized linear mixed effects model of the influence of month, diel period, and lunar phase on the probability of non-zero movement rates.**

**Model Equation: Binary Movement ~ Diel Period + Month + (1|Tagged Individual)**

**Fixed effects**

| | Estimate | SE | z value | Pr(>|z|) |
|---|---|---|---|---|
| Intercept | 0.784 | 0.242 | 3.236 | 0.001 |
| Diel period-Day | 0.436 | 0.102 | 4.264 | <0.001 |
| Diel period-Dusk | −0.310 | 0.134 | −2.309 | 0.021 |
| Diel period-Night | −0.422 | 0.104 | −4.054 | <0.001 |
| lunar phase-Waxing | −0.154 | 0.063 | −2.468 | 0.014 |
| lunar phase-Full | 0.098 | 0.065 | 1.517 | 0.129 |
| lunar phase-Waning | −0.070 | 0.063 | −1.108 | 0.268 |
| Month-2 | 0.276 | 0.092 | 3.011 | 0.003 |
| Month-3 | −0.382 | 0.086 | −4.468 | <0.001 |
| Month-4 | −0.167 | 0.091 | −1.826 | 0.068 |
| Month-5 | −0.752 | 0.128 | −5.899 | <0.001 |
| Month-6 | −1.033 | 0.139 | −7.421 | <0.001 |
| Month-7 | −0.831 | 0.144 | −5.768 | <0.001 |
| Month-8 | −1.097 | 0.122 | −8.963 | <0.001 |
| Month-9 | −1.468 | 0.131 | −11.168 | <0.001 |
| Month-10 | −0.600 | 0.115 | −5.218 | <0.001 |
| Month-11 | −0.404 | 0.092 | −4.388 | <0.001 |
| Month-12 | −0.179 | 0.087 | −2.059 | 0.040 |

Conditional $R^2$: 0.141 Marginal $R^2$: 0.083

**Note:**
The $p$ values shown were estimated based on Satterthwaite's approximation of degrees of freedom method (*Kuznetsova, Brockhoff & Christensen, 2017*). Conditional and marginal pseudo-$R^2$ values were estimated using the delta method (*Nakagawa, Johnson & Schielzeth, 2017*).

**Table 3 Summary of the linear mixed-effects model of the influence of month, diel period, and lunar phase on the non-zero movement rates (m/hr).**

**Model Equation: Non-Zero Movement Rates ~ Diel Period + Month + (1|Tagged Individual)**

**Fixed effects**

| | Estimate | SE | df | t value | Pr(>|t|) |
|---|---|---|---|---|---|
| Intercept | 639.948 | 43.696 | 19.405 | 14.646 | <0.001 |
| Diel period-Day | 11.786 | 29.294 | 5,047.145 | 0.402 | 0.687 |
| Diel period-Dusk | 5.134 | 39.558 | 5,050.903 | 0.130 | 0.897 |
| Diel period-Night | −77.660 | 29.869 | 5,051.998 | −2.600 | 0.009 |
| Lunar phase-Waxing | −9.374 | 17.822 | 5,052.914 | −0.526 | 0.599 |
| Lunar phase-Full | 7.420 | 18.055 | 5,052.095 | 0.411 | 0.681 |
| Lunar phase-Waning | 17.691 | 17.885 | 5,052.540 | 0.989 | 0.323 |
| Month-2 | 87.103 | 23.981 | 5,050.360 | 3.632 | <0.001 |
| Month-3 | 75.172 | 24.355 | 4,883.729 | 3.086 | 0.002 |

**Model Equation: Non-Zero Movement Rates ~ Diel Period + Month + (1|Tagged Individual)**

**Fixed effects**

| | Estimate | SE | df | t value | Pr(>|t|) |
|---|---|---|---|---|---|
| Month-4 | 112.366 | 25.028 | 4,587.245 | 4.490 | <0.001 |
| Month-5 | 143.228 | 39.229 | 5,033.253 | 3.651 | <0.001 |
| Month-6 | 201.114 | 42.854 | 4,977.599 | 4.693 | <0.001 |
| Month-7 | 249.189 | 41.064 | 4,980.760 | 6.068 | <0.001 |
| Month-8 | 61.872 | 37.665 | 5,050.979 | 1.643 | 0.101 |
| Month-9 | 30.634 | 44.148 | 4,991.306 | 0.694 | 0.488 |
| Month-10 | 36.870 | 33.904 | 5,001.054 | 1.087 | 0.277 |
| Month-11 | −33.524 | 26.195 | 5,023.740 | −1.280 | 0.201 |
| Month-12 | −19.478 | 24.391 | 5,051.994 | −0.799 | 0.425 |

Conditional $R^2$: 0.051 Marginal $R^2$: 0.033

**Note:**
The $p$ values shown were estimated based on based on Satterthwaite's degrees of freedom method (*Kuznetsova, Brockhoff & Christensen, 2017*). Conditional and marginal pseudo-$R^2$ values were estimated using the delta method (*Nakagawa, Johnson & Schielzeth, 2017*).

present in the La Jolla array, we detected them more often outside of the boundaries of local MPAs, in both spawning and non-spawning season, particularly in highly trafficked recreational fishing areas. While this could be an artifact of where we captured individuals (outside MPAs), it still indicates that Giant Sea Bass in La Jolla are at risk of the potential negative impacts of incidental recreational catch. Despite only gathering data on five individuals, our study offers valuable insight into the spatial ecology of this species, especially considering the length of time individuals were monitored and the paucity of published literature on this species in general.

Our results suggest that fish in the La Jolla array occupied relatively small, well-defined areas. Fish frequently did not move enough across hour time periods to be detected on multiple receivers (as evidenced by the zero-inflation of our movement rate data). Even the fish that traveled to Del Mar, while being detected on multiple receivers spanning the full extent of the array during it's time in La Jolla, showed remarkably consistent detections (averaging over 60 detections per hour) at the Del Mar receiver for a period of 5 months. The consistency of the detections was such that we initially considered that this was an incidence of mortality or tag expulsion near the receiver. After continued monitoring, the variability in detections indicating departure from and return to the area led us to believe the fish was alive and just consistently occupying the area near the Del Mar receiver. Furthermore, most of our tagged fish had the greatest detection rates on receivers near where they were captured. We conducted our tagging efforts outside of MPAs, thus most of our fish tended to be detected in areas outside of spatial protection. Somewhat counterintuitively, this trend of high site fidelity across smaller scales suggests that spatial management such as MPAs could be an effective tool for sheltering some individuals from fishing activity if their range is within the MPA, though more study is warranted to confirm this notion given our small sample size.
Our finding of high site fidelity for most individuals agrees with the findings of previous studies on Giant Sea Bass and similar species (*Eklund & Schull, 2001*; *Hanan & Curry, 2012*; *Clua et al., 2015*; *Spector et al., 2022*). Goliath Grouper (*Epinephelus itajara*) and Giant Grouper (*Epinephelus lanceolatus*), also show site fidelity across years, in some cases with individuals being resighted in the same location up to 4 years after the initial record (*Eklund & Schull, 2001*; *Giglio, Adelir-Alves & Bertoncini, 2014*; *Clua et al., 2015*). In La Jolla, it is possible the large spatial extent of contiguous kelp forest habitat and its ability to support ample prey resources contribute to the high site fidelity we observed among tagged Giant Sea Bass (*Parnell et al., 2006*; *Udy et al., 2019*). If site fidelity is driven by the availability of resources, it is possible it will lower as Giant Sea Bass populations continue to recover and intra-species competition for resources becomes more influential (*Atwell, O'Neal & Ketterson, 2011*; *Dmitrieva et al., 2016*).

We did not observe long-distance movements in our study, but it is possible these events occurred and went undetected given the sparse regional receiver coverage. Most fish disappeared from the La Jolla array prior to the end of tag battery life. This could be the result of either mortality outside of the receiver array or relocation to other areas. For the latter scenario, with the exception of the fish that went to Del Mar, we cannot say how far they could have traveled. Though we detected no fish on any of the regional receivers along the southern California coast, in the Channel Islands, or Baja California, Mexico, there is a chance that fish that left the La Jolla array made undetected long-distance excursions/relocations. Previous studies have documented such long-distance movements in tagged fish (both generally, and in the case of Giant Sea Bass). For instance, one of the species' congeners in the Polyprionidae family, the Hāpuku (*Polyprionidae oxygeneios*), showed variable movement patterns during a multi-year mark-recapture study with some being recaptured close to 1,400 km from their tagging location and others being recaptured at the same location as tagging (*Beentjes & Francis, 1999*). In another study, Giant Sea Bass tagged on Santa Catalina Island, California traveled long distances from the island following spawning season, traversing the San Pedro Channel to the mainland, or traveling to other islands in the area (*Burns et al., 2020*; *Clevenstine & Lowe, 2021*). Given these results from other studies, it is not out of the realm of possibility for individuals to travel from the San Diego region to areas where they are susceptible to either targeted or incidental commercial catch, such as Baja California, Mexico or outside of the 3-mile nearshore gill and trammel net ban. While we can't rule out such movements in our tagged fish, the relatively long residence of several individuals to La Jolla (three of seven fish detected in the region for ~2–4 years) suggests at least some Giant Sea Bass have strong site fidelity to a coastal region with intensive spatial fisheries restrictions.

We found that Giant Sea Bass tended to have a higher probability of movement during the day and that their movement patterns did not seem to be influenced by lunar phase. *Clevenstine & Lowe (2021)* found similar results on Santa Catalina Island with longer distances traveled on average during the day and no effect of lunar phase on movement during spawning season. We agree with their assessment that the lack of influence of lunar phase could be the result of abundances being too low to support consistent aggregation behavior, or alternatively, that Giant Sea Bass could be more akin to aggregation forming

species that do not align their spawning with particular lunar phases, such as Gulf Grouper (*Mycteroperca jordani*) (*Rowell et al., 2019*; *Clevenstine & Lowe, 2021*). We should note that our movement models indicated that our explanatory variables did not account for much of the variability in our data as evidenced by the low conditional $R^2$ values. We suspect they would have higher explanatory power if we had a more expansive receiver array and could better resolve the finer-scale movement patterns of our tagged fish. More research into the periodicity of Giant Sea Bass spawning behavior is warranted, especially as their populations continue to recover.

Spatial management tools would be most effective if they encompassed spawning aggregation sites for this fish. Previous studies showed that spatial protections of spawning aggregations can help support recovery from overfishing (*Nemeth, 2005*; *Chollett et al., 2020*; *Waterhouse et al., 2020*). While the California MPAs were not implemented with Giant Sea Bass in mind, if MPA boundaries include spawning aggregation sites they could help support the species' population recovery by protecting fish during a critical stage of their life history (*Chollett et al., 2020*). Our results suggest that there is likely a spawning aggregation in La Jolla—we detected fish year-round and found seasonal differences in movement during the presumed spawning season. Due to our small sample size and array positioning, we cannot say whether a La Jolla aggregation is inside or outside MPAs. However, fish were most active in the northwest corner of the array during the spawning season, which coincides with heavily trafficked fishing grounds. Previous characterizations of spawning aggregation sites of Giant Sea Bass and similar species occurring near promontories in areas with strong currents (*Eklund & Schull, 2001*; *Clevenstine & Lowe, 2021*) support the notion of an aggregation in this area. The La Jolla submarine canyon runs along the northwest corner of the La Jolla array and is home to steep sandstone cliffs and subsurface promontories that contribute to the generation of strong currents close to the edge of the kelp forest (*Parnell et al., 2005*, *2006*, *2010*). Incidentally, these same currents are responsible for attracting pelagic migratory species that are highly sought after by recreational anglers (*Parnell et al., 2010*).

We detected high levels of activity and evidence for a potential spawning aggregation in one of the most highly trafficked recreational fishing areas in San Diego (*Parnell et al., 2010*). While much of the local recreational fishing community is conscientious of regulations and efforts to support the recovery of Giant Sea Bass, fatalities do occur because of incidental catch. Barotrauma can occur when there is rapid change in pressure, such as when a fish is brought to the surface quickly from depth, resulting in an overexpansion of gases in the body of the fish, especially in the swim bladder (*Rummer & Bennett, 2005*; *Parker et al., 2006*; *Jarvis & Lowe, 2008*). Giant Sea Bass are susceptible to barotrauma and, as large animals, can be difficult to handle properly. One of the strongest indicators of post-release survival following barotrauma in other species is the ability to release the fish as quickly as possible (*Jarvis & Lowe, 2008*; *Roach, Hall & Broadhurst, 2011*). With a fish that regularly reaches over a meter in length and is often interacting with anglers on kayaks or larger chartered fishing vessels with raised decks (*Parnell et al., 2010*), reducing surface time is especially challenging. In the event a fish is released successfully, there is still a chance delayed mortality can occur if there is excessive damage to the swim

bladder or other organs (*Parker et al., 2006*; *Jarvis & Lowe, 2008*). Furthermore, sublethal effects of catch and release fishing can also negatively impact individuals by decreasing their overall fitness (*Cooke & Schramm, 2007*; *Campbell et al., 2010*).

While incidental catch is of concern, the magnitude of negative effects is not so strong that it hindered the ongoing recovery of Giant Sea Bass in recent years. Our finding that individuals persisted in La Jolla across multiple years suggests the existing spatial and fisheries management measures afford protection to Giant Sea Bass in the area. Management of the ongoing recovery of this species throughout its range would benefit from further work quantifying the effects of incidental recreational catch. Fortunately, there is an understanding of best practices to mitigate the effects of incidental catch, chief among them quickly and efficiently releasing fish back to depth. Development of tools, such as larger versions of descending devices (*e.g.*, SeaQualizers) often used with rockfish, can help support efforts to properly handle incidentally caught Giant Sea Bass.

## CONCLUSIONS

We used acoustic telemetry to characterize the residency and seasonality of Giant Sea Bass space use in one of the largest kelp forests in southern California and identified how tagged individuals interact with local spatial management and fishing activity. While our sample size was small (five fish), we found valuable information on this Critically Endangered species. We found fish were resident to the La Jolla area for extended periods, with the longest consistent detection range lasting 4 years. We are unsure where the fish traveled after leaving the array, as we detected only one fish on receivers maintained in the broader southern California and Baja California region. Receiver coverage was sparse in many areas, especially in Mexican waters. Fish were detected less frequently and displayed higher movement rates during spawning months. Based on the movement patterns, tagged fish are regularly interacting with a highly trafficked recreational fishing ground, including during spawning season. Nevertheless, existing spatial and fishery management measures appear to support long-term persistence of Giant Sea Bass in an area that is marked by high recreational fishing pressure.

## ACKNOWLEDGEMENTS

We acknowledge all the other regional acoustic telemetry research groups that queried their databases for our tag numbers, including Brian Sterling and Chris Lowe from the Lowe Lab at CSULB, Ryan Freedman and Pike Spector from the Channel Islands National Marine Sanctuary, and James Ketchum, Marc Aquino Baleytó, Mauricio Hoyos from Pelagios-Kakunjá A.C. in Baja California, Mexico. We thank the Hubbs Sea World Research Institute for allowing us to follow them during their surveys in case they caught a Giant Sea Bass. We would like to thank Phil Zerofski and Chugey Sepulveda for helping us develop our fishing and tagging protocols. We would also like to acknowledge the *many* volunteer anglers and divers who helped with the field work for this project, especially Rich Walsh, Ross Cooper, Zach Skelton, Erica Jarvis-Mason, Shane Finnerty, Mohammad Sedarat, and Youssef Doss.

### Funding
The following funding sources contributed to this work: National Marine Fisheries Service, NOAA, Quantitative Ecology and Socioeconomics Training (QUEST) Program, the Mia Tegner Memorial Fellowship, the SIO Center for Marine Biodiversity and Conservation Mentorship Program, the Women Divers Hall of Fame Marine Conservation Scholarship sponsored by the Rachel Morrison Memorial Fund, and the Link Family Foundation (*via* Dr. Phil Hastings). The funders had no role in study design, data collection and analysis, decision to publish, or preparation of the manuscript.

### Grant Disclosures
The following grant information was disclosed by the authors:
National Marine Fisheries Service.
NOAA.
Quantitative Ecology and Socioeconomics Training (QUEST) Program.
Mia Tegner Memorial Fellowship.
SIO Center for Marine Biodiversity and Conservation Mentorship Program.
Women Divers Hall of Fame Marine Conservation Scholarship sponsored by the Rachel Morrison Memorial Fund.
Link Family Foundation.

### Competing Interests
The authors declare that they have no competing interests.

### Author Contributions
- Kayla M. Blincow conceived and designed the experiments, performed the experiments, analyzed the data, prepared figures and/or tables, authored or reviewed drafts of the article, and approved the final draft.
- Jack T. Elstner performed the experiments, authored or reviewed drafts of the article, and approved the final draft.
- Noah Ben-Aderet conceived and designed the experiments, authored or reviewed drafts of the article, and approved the final draft.
- Lyall F. Bellquist conceived and designed the experiments, authored or reviewed drafts of the article, and approved the final draft.
- Andrew P. Nosal performed the experiments, authored or reviewed drafts of the article, and approved the final draft.
- Brice X. Semmens conceived and designed the experiments, authored or reviewed drafts of the article, and approved the final draft.

### Animal Ethics
The following information was supplied relating to ethical approvals (*i.e.*, approving body and any reference numbers):

The following International Animal Care and Use Committees approved this research: University of California, San Diego IACUC (Protocol #S12116) and Hubbs Sea World Research Institute IACUC (Protocol APF #2016-09).

## Field Study Permissions

The following information was supplied relating to field study approvals (*i.e.*, approving body and any reference numbers):

California Department of Fish and Wildlife approved all collections (Permit #S-192900002-19290-001) and (Permit: P1770011).

## Data Availability

The data and code to perform analyses is available at Zenodo: Kayla. (2023). kmblincow/GSBMovement_MS: GSBMovementPublication_ReviewerComments (v1.1). Zenodo. https://doi.org/10.5281/zenodo.10067113.

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
