# Peer review of "Spatial ecology of the Giant Sea Bass, Stereolepis gigas, in a southern California kelp forest as determined by acoustic telemetry"

_PeerJ, doi:10.7717/peerj.16551_

## Round 0.1 · original submission · Major Revisions

Your manuscript has been reviewed by three referees. Two of them requested revision (and a third one provided suggestions also but these were minimum). I agree with the suggestions provided by all reviewers to clarify many issues addressed in the manuscript. Consequently, I kindly invite you to address all corrections please and respond accordingly.

Reviewer 1 ·

Basic reporting

This article meets PeerJ standards. The introduction section is a bit choppy - sentences are short and not woven together as well as the rest of the paper. Not sure if it's worth adding more information or combining ideas to improve it. The materials and methods section is very well written - clear, concise, and understandable to a wide range of audiences.

Experimental design

This article meets PeerJ standards. The sample size is quite small (five individuals) but the methods used to asses the hypothesis are sound and follow technical and ethical standards. Please include receiver depth range for the entire array and for those included in the detection range analysis. If pertinent, how could depth have impacted detectability, particularly in the northwest portion of the array where it's noted current can be strong.

Validity of the findings

The article meets PeerJ standards. Overall conclusions are well stated but would benefit from more information on how such a small sample size may impact their findings and overall conclusions. Lines 338-339: Examples like Rowell et al. (2019) and papers cited herein (Clevenstine & Lowe 2021, Spector et al. 2022), found substantially higher movement rates at night during the spawning season. Although this array did not encompass a presumed spawning location what can you infer about this difference in diel movement? Line 392: Could the detections of 60/min be because the fish died or the tag was expelled? Line 397: With such a small sample size consider toning this down or explaining what it means in light of the number of tagged individuals.

Additional comments

This article is very well written, for the most part, and scientifically sound. It must be noted that, because of the limited sample size, the paper needs to be just a bit more forthright about the limitations of conclusions. I think this paper would benefit from an exploration on how climate change and changing oceans may impact this species and existing MPAs or restricted use areas. How can what has been presented here be used to improve spatial management for other species in the eastern Pacific or elsewhere? How might recreational fishing pressure in the US change as spatial management and species distribution evolves? Line 398-399: This is a great point that could be explored a bit more. Line 421: I think this needs to be woven into the narrative a bit more, considering only five individuals were included and one moved out of the array for the rest of the study. Although there is ample data for the fish that remained, I think the very small sample size needs to be reiterated and a little bit more thoroughly discussed. Line 435: Can you infer transit speed between La Jolla and Del Mar? Line 459: Not sure if "substantial" is the best choice here, considering you mentioned tagged individuals spent much of their time outside of the array.

Reviewer 2 ·

Excellent Review

This review has been rated excellent by staff (in the top 15% of reviews)
EDITOR COMMENT
It was a detailed and constructive review. Thank you for your very kind attention to this review.

Basic reporting

Please increase the font size of figure text and consider including color in figures (other than fig. 4) to increase easy of interpreting them.

line 511: This literature reference is missing data.

line 545: This literature reference is missing data.

Experimental design

I would like to have a little more information added about the fish that was tagged on the boat rather than alongside the boat in the water. It sounds like the fish was tagged out of the water with no flow through and just wet towels over it’s head. How long was the fish out of the water and which fish was it? One of the fish that had a good amount of detections, or one that was never heard from? The permits and IACUC approval all seems to be in place, I am just surprised at the method of tagging for that one fish and thought that IACUC still required anesthesia to be used during tagging procedures. At the very least, please add some justification why a large tank or hose with flow through water was not used.

Also, I do not see depths of receivers listed anywhere and there are no contour lines on the study area map. Add depths and if possible, the multibeam data you mentioned, to better show the area and where these fish may have moved to when they went out of the array.

Validity of the findings

Having data on only 5 fish (with only 2 of them having multiple years of data), is a very small sample size to make generalizations about population level threats and spatial management, and is also very low to discuss spawning behavior. I would be more cautious than you currently are in how you connect the movements of 5 fish in making inferences in the discussion and conclusions.

Although 5 of the 7 fish were detected for a relatively long time in the open to fishing area, most stopped getting detected long before their tag expiration dates. Did you look into the last receivers these fish were detected on? Were they still in the open fish area? Were they all on the deeper edge of the array, indicating that fish moved into deeper water? Or was it likely incidental catch or predation? Also, please address if all fish are likely sexually mature before going into movements during the spawning months. Fish 1, with the longest dataset, seems like it may not be mature based on House et al. 2016 and Hawk and Allen 2014.

If you wanted to look at how movement relates to the spawning season why not include lunar phase along with month and diel period? Spawning behaviors in fish have a strong relationship with the lunar phase and it is fairly easy to calculate the lunar phase based on date and include in the GLMM.

It is also mentioned that the fish don't appear to use the MPAs very much, but there looks to be big chunks of the MPAs that do not have receiver coverage, so careful how you phrase things in terms of them not using the MPAs.

Additional comments

Overall the manuscript is well-written, with clear language. However, there are areas where some restructuring would improve the reading.

In my first read of this manuscript, I often read something that needed more explanation or follow up, and would make a note of it, only for that follow up to be included later than I expected. For example, when discussing the fact that most fish were outside of the array in summer spawning months (line ~354), the follow up to explain where they might be going based on bathymetry and spawning needs occurred more than 60 lines later (line ~416). Similarly, line 421 mentions that the results provide insight into the susceptibility of fish to their three major fishing-related threats, the presentation of recreational fishing as the greatest threat isn't directly stated until the next paragraph. I suggest some editing to lead right off the bat with your most compelling finding, then go into why this is, at both the sentence and paragraph level. There are times where the secondary details or prepositional phrases come first and the major point follows later, losing some of it's impact.

The methods are well explained on how and why the study was done and the introduction does a good job of explaining about the gap of knowledge in giant sea bass spatial ecology.


line 152: I don't think interact is the right word here. "how tagged individuals are affect by spatial management and...."? Respond to?

line 158: rephrase slightly. Maybe "...and bounded on the western and southern edges by sandy bottom habitats.

line 173: Did you plan to only tag 7 fish or were there limitations? Permit limitations or a lack of fish would be an easy thing to incorporate, but some explanation somewhere of the low sample size would help.

line 243: how far apart were receivers? Were they close enough for the same ping to be detected by multiple receivers and if so, did you remove the later detection on the other receiver? Since you used center of activities, that does not seem necessary and may contribute to more zero-movement events.

line 245: Did you filter out any single detections within a certain time frame? I'm trying to understand how fish #2 had 600+ days at liberty, but the plot looks like it had less than a year of detections.

line 305: How much larger was the detection range in the sandy habitat? Can you provide the average in the kelp forest and in sand areas? It would be helpful for other researchers to have an idea of how different the detections range is between these two habitats. Is there supplementary material to cite about the difference between ranges? Or plot both kelp bed and sand detection range curves along with the averages?

line 309: You describe what time at liberty in the array means in the table captions, but I don't see it in the text before this. Here, you use the terms study residency index and array residency index. Remain consistent so readers don't have to search for meanings.

line 342: I would guess that lunar phase and the interaction of lunar phase and diel period would explain more of the variance. Potentially temperature too, though that may be partially explained with month as a factor. I may have missed it, but I don't see you address the other possible sources of the movement variance in the discussion.

line 355: Where do you think they went? A lot spawning fish will gather at promontories at deeper ledges. Do you suspect they went farther west to deeper water? What were the last receivers to detect fish as they left the array?

line 357: Add here that they are still within the 3 mile gear ban and the threat would be from incidental recreational catch, if that is the case.

line 397: Fish may not necessarily be using unprotected habitat over MPAs, but there may be a territorial effect where the fish in the MPAs stay there most of the time. It would have been interesting to tag inside and outside of the MPAs to look at differences in residency. I think you move this sentence up to the previous paragraph where you state that the fish spend most of their time outside of MPAs.

line 401: Remove the word 'likely'

line 479: You stated before that it was likely there was a spawning aggregation in the area, but sections of the conservation areas don't have receivers and from the scale bar it looks like several kms inside the no-take areas don't have coverage. I would state that more work is needed to determine if a spawning aggregation is located inside an MPA.

Table 1: Fix the column headings/be sure to check your proofs, they came out wonky. Is size at maturity known for giant sea bass? Fish 1 appears to be a juvenile based on size and House et al. 2016 and Hawk and Allen 2014. And how does fish #2 have 623 days at liberty when the plot looks like it has less than a year of data?

Figure 1: I would make the extent box in the zoomed out map thicker, and ideally a different color. Add lines from that extent box to the inset map to make it clear where those locations are. If this figure is displayed smaller, it is going to be tough to see.

I also don't really like how the protected areas are described as to the north and south when it isn't clear where that reference point is. If I understand correctly, La Jolla is that whole general region. Also, the areas you describe as to the south extend to the west from the coast, where as the areas to the north extend north. That may be another reason the description bothers me. Maybe describe those protected areas as covering the northern or southern part of your la jolla array?

Can you add depth contour lines? I have no idea what the depths are.

Figure 2: I suggest making the symbols different colors. It is difficult to separate out the line of averages versus the La jolla detections, and if there are del mar detections before august/sept time frame, it is almost impossible to see them.

Figure 3: Odd code at the end of this caption and at the end of figure 5's caption. Again, make sure you go through your proofs so they don't get published like this.

Figure 4: I suggest separating B-F to be it's own figure. the plots are small, and the text too small to read without zooming in 150%. Please make the plots larger and/or the text larger. For plot A: any ideas on why there is an almost season pattern for the smallest fish's detections? Some years it is present for the majority of summer months, other years it looks to be more absent during summers. You could focus more on the inter-variability of these fish.... Fish 7 also spends most of it's time in an MPA, but was tagged outside of it....Is the variation in space use and presence of fish in the array potentially due to size or sex? I think that may be beneficial to mention, if there is literature to support size or sex based differences in giant sea bass, do these patterns match?

Figure 5: Similar to other figures, make the text larger, and add color to make them easier to read.

Reviewer 3 ·

Basic reporting

The manuscript is very comprehensive and well written.

Experimental design

The experimental design and analyses are robust and thoroughly described, mentioning the limited sample size (n = 5 with data).

Validity of the findings

The conclusions drawn from the data are appropriate and the manuscript overall reads very well with logical structure.

Additional comments

I have only a couple of very minor comments that should be addressed prior to publishing, though they are so minor that I would still recommend accepting the manuscript as is.

Materials & Methods

Lines 209 & 464: “SeaQualizer” (seaqualizer.com) is misspelled.

Results

Line 319-320: Can you provide any habitat characteristics for the Del Mar receiver station?

---

## Round 0.2 · Minor Revisions

I reviewed the last version and found various details in need to be addressed. You will receive the Word document to be easy to follow, but I paste a PDF in the meantime.

Reviewer 2 ·

Basic reporting

The writing in this revised manuscript is much clearer and cleaner. It tells a good story, describes the methods well, and clarifies some points that were unclear in the previous version. The figures are also much improved. The addition of color and increase of font size makes the figures much easier to interpret and are also nicer to look at. The addition of depth in the location map is also very helpful.

Experimental design

The revised version clarified the questions I had about the methods. The addition of lunar phase in the model is appreciated, and the fact that it didn't explain much more of the variance is interesting. The authors addressed this and other aspects with sufficient detail. The addition of the habitat curves in the detection range is valuable and adds useful details for others who use telemetry in the area.

Validity of the findings

The inclusion of more caveats about the results is appreciated since there were only 5 fish who had movement data. While the results here do show longer detection data than previous studies, the authors rewording to describe the data in terms of individuals rather than the population level is appropriate.

Additional comments

The authors have done a great job addressing reviewer concerns and as a result, this manuscript is much improved and is ready to be accepted.

---

## Round 0.3 · accepted · Accept

You have fulfilled the recommendations properly.